# IMPROVING CALIBRATION THROUGH THE RELATIONSHIP WITH ADVERSARIAL ROBUSTNESS

## ABSTRACT

Neural networks lack *adversarial robustness* – they are vulnerable to adversarial examples that through small perturbations to inputs cause incorrect predictions. Further, trust is undermined when models give *miscalibrated* uncertainty estimates, i.e. the predicted probability is not a good indicator of how much we should trust our model. In this paper, we study the connection between adversarial robustness and calibration on four classification networks and datasets. We find that the inputs for which the model is sensitive to small perturbations (are easily attacked) are more likely to have poorly calibrated predictions. Based on this insight, we examine if calibration can be improved by addressing those adversarially unrobust inputs. To this end, we propose Adversarial Robustness based Adaptive Label Smoothing (`AR-AdaLS`) that integrates the correlations of adversarial robustness and uncertainty into training by adaptively softening labels for an example based on how easily it can be attacked by an adversary. We find that our method, taking the adversarial robustness of the in-distribution data into consideration, leads to better calibration over the model even under distributional shifts. In addition, `AR-AdaLS` can also be applied to an ensemble model to further improve model's calibration.

## 1 INTRODUCTION

The robustness of machine learning algorithms is becoming increasingly important as ML systems are being used in higher-stakes applications. In one line of research, neural networks are shown to lack *adversarial robustness* – small perturbations to the input can successfully fool classifiers into making incorrect predictions (Szegedy et al., 2014; Goodfellow et al., 2014; Carlini & Wagner, 2017b; Madry et al., 2017; Qin et al., 2020b). In largely separate lines of work, researchers have studied uncertainty in model's predictions. For example, models are often *miscalibrated* where the predicted confidence is not indicative of the true likelihood of the model being correct (Guo et al., 2017; Thulasidasan et al., 2019; Lakshminarayanan et al., 2017; Wen et al., 2020; Kull et al., 2019). The calibration issue is exacerbated when models are asked to make predictions on data different from the training distribution (Snoek et al., 2019), which becomes an issue in practical settings where it is important that we can trust model predictions under distributional shift.

Despite robustness, in all its forms, being a popular area of research, the *relationship* between these perspectives has not been extensively explored previously. In this paper, we study the correlation between adversarial robustness and calibration. We discover that input data that are sensitive to small adversarial perturbations (are easily attacked) are more likely to have poorly calibrated predictions. This holds true on a number of network architectures for classification and on all the datasets that we consider: SVHN (Netzer et al., 2011), CIFAR-10 (Krizhevsky, 2009), CIFAR-100 (Krizhevsky, 2009) and ImageNet (Russakovsky et al., 2015). This suggests that the miscalibrated uncertainty estimates on those adversarially unrobust data greatly degrades the performance of a model's calibration. Based on this insight, we hypothesize and study if calibration can be improved by giving different supervision to the model depending on adversarial robustness of each training data.

To this end, we propose **A**dversarial **R**obustness based **Ada**ptive **L**abel **S**moothing (`AR-AdaLS`) that integrates the correlations of adversarial robustness and calibration into training by adaptively smoothing the training labels conditioned on how unrobust an input is. Our method improves label smoothing (Szegedy et al., 2014) by explicitly teaching the model to differentiate the training data according to their adversarial robustness and then adaptively smooth their labels. By giving different

supervision to the training data, our method leads to better calibration over the model without an increase of latency during inference. In particular, since adversarially unrobust data can be considered as an outlier of the underlying data distribution (Carlini et al., 2019), our method, by taking the adversarial robustness of the in-distribution data into consideration during training, can even greatly improve model's calibration on held-out shifted data. Further, we propose "AR-AdaLS of Ensemble" to combine our AR-AdaLS and deep ensembles (Lakshminarayanan et al., 2017), which is the state-of-the-art method especially under distributional shift (Snoek et al., 2019), to further improve the calibration performance for shifted data. Last, we find an additional benefit of AR-AdaLS is improving model stability (i.e., decreasing variance), which is valuable in practical applications where changes in predictions across runs (churn) is problematic and deploying ensembles is too costly (Milani Fard et al., 2016).

In summary, our main contributions are as follows:

- **Relationship among Robustness Metrics:** We find a significant correlation between adversarial robustness and calibration: inputs that are unrobust to adversarial attacks are more likely to have poorly calibrated predictions.
- **Algorithm:** We propose AR-AdaLS to automatically learn how much to soften the labels of training data based on their adersarial robustness. Further, we introduce "AR-AdaLS of Ensemble" to show how to apply AR-AdaLS to an ensemble model.
- **Experimental Analysis:** On CIFAR-10, CIFAR-100 and ImageNet, we find that AR-AdaLS is more effective than previous label smoothing methods in improving calibration, particularly for shifted data. Further, we find that while ensembling can be beneficial, applying AR-AdaLS to adaptively calibrate ensembles offers further improvements over calibration.

## 2 RELATED WORK

**Uncertainty Estimates**  How to better estimate a model's predictive uncertainty is an important research topic, since many models with a focus on accuracy may fall short in predictive uncertainty. A popular way to improve a model's predictive uncertainty is to make the model well-calibrated, e.g., post-hoc calibration by temperature scaling (Guo et al., 2017), and multiclass Dirichlet calibration (Kull et al., 2019). In addition, Bayesian neural networks, through learning a posterior distribution over network parameters, can also be used to quantify a model's predictive uncertainty, e.g., Graves (2011); Blundell et al. (2015); Welling & Teh (2011). Dropout-based variational inference (Gal & Ghahramani, 2016; Kingma et al., 2015) can help DNN models make less over-confident predictions and be better calibrated. Recently, mixup training (Zhang et al., 2018) has been shown to improve both models' generalization and calibration (Thulasidasan et al., 2019), by preventing the model from being over-confident in its predictions. Despite the success of improving uncertainty estimates over in-distribution data, Snoek et al. (2019) argue that it does not usually translate to a better performance on data that shift from the training distribution. Among all the methods evaluated by Snoek et al. (2019) under distributional shift, ensemble of deep neural networks (Lakshminarayanan et al., 2017), is shown to be most robust to dataset shift, producing the best uncertainty estimates.

**Adversarial Robustness**  On the other hand, machine learning models are known to be brittle (Xin et al., 2017) and vulnerable to adversarial examples (Athalye et al., 2018; Carlini & Wagner, 2017a;b; He et al., 2018). Many defenses have been proposed to improve model's adversarial robustness (Song et al., 2017; Yang et al., 2019; Goodfellow et al., 2018), however are further attacked by more advanced defense-aware attacks (Carlini & Wagner, 2017b; Athalye et al., 2018). Recently, Carlini et al. (2019); Stock & Cissé (2018) define adversarial robustness as the minimum distance in the input domain required to change the model's output prediction by constructing an adversarial attack. The most recent work that is close to ours, Carlini et al. (2019), makes the interesting observation that easily attackable data are often outliers in the underlying data distribution and then use adversarial robustness to determine an improved ordering for curriculum learning. Our work, instead, explores the relationship between adversarial robustness and calibration. In addition, we use adversarial robustness as an indicator to adaptively smooth the training labels to improve model's calibration.

**Label Smoothing**  Label smoothing is originally proposed in Szegedy et al. (2016) and is shown to be effective in improving the quality of uncertainty estimates in Müller et al. (2019); Thulasidasan

Table 1: Network architecture and accuracy used for each dataset.

| Dataset | SVHN | CIFAR-10 | CIFAR-100 | ImageNet |
|---------|------|----------|-----------|----------|
| Network | CNN-7 | ResNet-29 v2 | Wide ResNet-28-10 v2 | ResNet-101 v1 |
| Accuracy | 95.0% | 91.4% | 79.2% | 77.7% |

et al. (2019). Instead of minimizing the cross-entropy loss between the predicted probability $\hat{p}$ and the one-hot label $p$, label smoothing minimizes the cross-entropy between the predicted probability and a softened label $\widetilde{p} = p(1 - \epsilon) + \frac{\epsilon}{Z}$, where $Z$ is the number of classes in the dataset and $\epsilon$ is a hyperparameter which controls the degree of the smoothing effect. Our work makes label smoothing adaptive and incorporates the correlation with adversarial robustness to further improve calibration.

## 3 CORRELATIONS BETWEEN ADVERSARIAL ROBUSTNESS AND CALIBRATION

To explore the relationship between adversarial robustness and calibration, we first introduce the metrics to evaluate each of them. We use arrows to indicate which direction is better.

**Adversarial robustness** $\uparrow$   Adversarial robustness measures the minimum distance in the input domain required to change the model's output prediction by constructing an adversarial attack (Carlini et al., 2019; Stock & Cissé, 2018). Specifically, given an input $x$ and a classifier $f(\cdot)$ that predicts the class for the input, the adversarial robustness is defined as the minimum adversarial perturbation $\delta$ that enables $f(x + \delta) \neq f(x)$. Following the work (Carlini et al., 2019), we construct the $\ell_2$ based CW attack (Carlini & Wagner, 2017b) and then use the $\ell_2$ norm of the adversarial perturbation $\|\delta\|_2$ to measure the distance to the decision boundary. Therefore, a more adversarially robust input requires a larger adversarial perturbation to change the model's prediction.

**Calibration metric** $\downarrow$   Model's calibration measures the alignment between the predicted probability and the accuracy. Well calibrated uncertainty estimates convey the information about how much we should trust a model's prediction. We follow the widely used expected calibration error (**ECE**) to measure the calibration performance of a network (Guo et al., 2017; Snoek et al., 2019). To compute the ECE, we need to first divide all the data into $K$ buckets sorted by their predicted probability (confidence) of the predicted class. Let $B_k$ represent the set of data in the $k$-th confidence bucket. Then the accuracy and the confidence of $B_k$ are defined as $\mathrm{acc}(B_k) = \frac{1}{|B_k|}\sum_{i \in B_k}\mathbf{1}(\hat{y}_i = y_i)$ and $\mathrm{conf}(B_k) = \frac{1}{|B_k|}\sum_{i \in B_k}\hat{p}_i^{\hat{y}_i}$, where $\hat{y}$ and $y$ represent the predicted class and the true class respectively, and $\hat{p}^{\hat{y}}$ is the predicted probability of $\hat{y}$. The ECE is then defined as $\mathrm{ECE} = \sum_{k=1}^{K}\frac{|B_k|}{N}|\mathrm{acc}(B_k) - \mathrm{conf}(B_k)|$, where $N$ is the number of the data.

### 3.1 CORRELATIONS

In this section, we perform experiments on the clean test set across four datasets: SVHN (Netzer et al., 2011), CIFAR-10 (Krizhevsky, 2009), CIFAR-100 (Krizhevsky, 2009) and ImageNet (Russakovsky et al., 2015) with different networks, whose architecture and accuracy are shown in Table 1. We refer to these models as "Vanilla" for each dataset in the following discussion. The details for training each vanilla network are included in Section A in the Appendix.

To explore the relationship between adversarial robustness and calibration, we start with the relationship between adversarial robustness and confidence together with accuracy. Specifically, we rank the input data according to their adversarial robustness and then divide the dataset into $R$ equally-sized subsets ($R = 10$ used in this paper). For each adversarial robustness subset, we compute the accuracy and the average confidence score of the predicted class. As shown in the first row in Figure 1, we can clearly see that both accuracy and confidence increase as adversarial robustness of the input data, and confidence is consistently higher than accuracy in each adversarial robustness subset across four datasets. This indicates that although vanilla classification models achieve the state-of-the-art accuracy, they tend to give over-confident predictions, especially for those unrobust data.

Based on this, to explore the relationship between adversarial robustness and calibration, we compute the expected calibration error (ECE) in each adversarial robustness subset, shown in the bottom row of Figure 1. In general, we find that those unrobust data in lower adversarial robustness levels are more likely to be over-confident and less well calibrated (larger ECE). For more robust examples, there is a better alignment between their confidence and accuracy, and the ECE over those examples is close to 0. On larger-scale ImageNet, while we still see the general trend that less robust examples

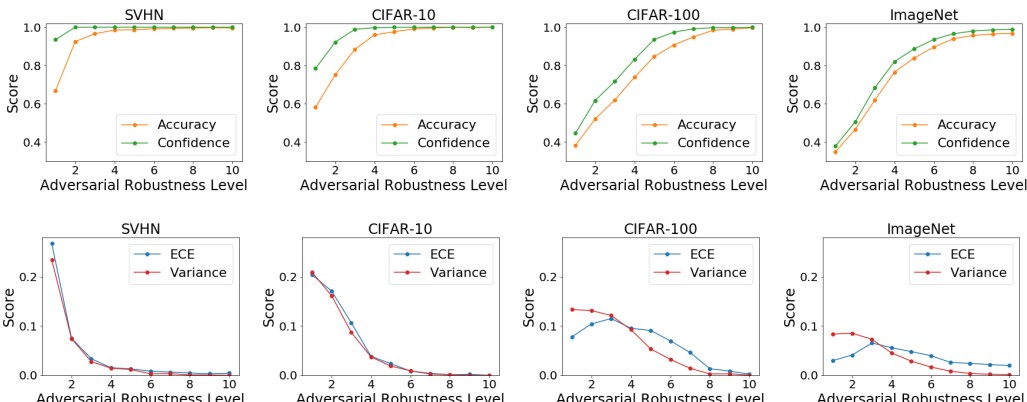

Figure 1: Correlations between adversarial robustness and uncertainty estimates on the clean test set on SVHN, CIFAR-10, CIFAR-100 and ImageNet. **Top:** Accuracy and confidence of the predicted class. **Bottom:** ECE and variance (lower is better)in each adversarial robustness subset. Higher adversarial robustness level means the input are more adversarially robust (harder to attack).

are less well calibrated, we see that the least robust examples are relatively well calibrated. We hypothesize this may be due to larger data and less overfitting.

Furthermore, we also find an interesting correlation between adversarial robustness and model stability, which is measured by the variance of the predicted probability across 5 independent runs. As shown in the bottom row of Figure 1, we see that those adversarially unrobust examples tend to have a much higher variance compared to the robust across all four datasets.

## 4 METHOD

Based on the correlation between adversarial robustness and calibration, we hypothesize and study if calibration can be improved by giving different supervision to the model depending on the adversarial robustness of training data. To this end, we propose a method named **A**dversarial **R**obustness based **Ada**ptive **L**abel **S**moothing (**AR-AdaLS**), which performs label smoothing at different degrees to the training data based on their adversarial robustness. Specifically, we sort and divide the training data into $R$ small subsets with equal size according to their adversarial robustness[1] and then use $\epsilon_r$ to soften the labels in each training subset $S_r^{train}$. The soft labels can be formulated as:

$$\widetilde{p}_r = p_r(1 - \epsilon_r) + \frac{\epsilon_r}{Z}, \tag{1}$$

where $p_r^{z=y} = 1$ for the correct class $y$ and $p_r^{z \neq y} = 0$ for the others, $p_r$ stands for the one-hot label, and $Z$ is the number of classes in the dataset. The parameter $\epsilon_r$ controls the degree of smoothing effect and allows for different levels of smoothing in each adversarial robustness subset. Generally, a relatively larger $\epsilon_r$ is desirable for lower adversarial robustness levels such that the model learns to make a lower confidence prediction. Instead of empirically setting the parameter $\epsilon_r$ in each adversarial robustness subset, we allow it to be adaptively updated according to the calibration performance on the validation set (discussed in Section 4.1). In this way, we explicitly train a network with different supervision based on the adversarial robustness of training data.

### 4.1 ADAPTIVE LEARNING MECHANISM

To find the best hyperparameter $\epsilon$ for label smoothing, previous methods (Szegedy et al., 2016; Thulasidasan et al., 2019) sweep $\epsilon$ in a range and choose the one that has the best validation performance. However, in our setting, the number of combinations of $\epsilon_r$ increases exponentially with the number of adversarial robustness subsets $R$. To this end, we propose an adaptive learning mechanism to automatically learn the parameter $\epsilon_r$ in each adversarial robustness subset. The overall training procedure is summarized in Algorithm 1.

First, we denote the soft label for the correct class in the $r$-th adversarial robustness subset as $\widetilde{p}_r^{z=y}$. According to Eqn. (1), we can derive:

---

[1]Note, predicted confidence is not a good indicator for splitting the training dataset as the model can easily overfit to the training data and their predicted confidence are all close to 100%.

---

**Algorithm 1** Pseudocode of the training procedure for `AR-AdaLS`

---

**Input:** number of classes $Z$, number of training epochs $T$, number of adversarial robustness subset $R$, learning rate of adaptive label smoothing $\alpha$.

For each adversarial robustness training subset, we initialize the soft label as the one-hot label $\widetilde{p}_{r,t} = p_r$, and initialize the soft label for the correct class $\widetilde{p}_{r,t}^{z=y} = 1$.

**for** $t = 1$ **to** $T$ **do**

    Minimize cross-entropy loss between soft label and predicted probability $\frac{1}{R}\sum_r^R \mathcal{L}(\widetilde{p}_{r,t}, \hat{p}_{r,t})$

    **for** $r = 1$ **to** $R$ **do**

        Update $\widetilde{p}_{r,t}^{z=y} \leftarrow \widetilde{p}_{r,t}^{z=y} - \alpha \cdot \{\mathrm{conf}(S_r^{val})_t - \mathrm{acc}(S_r^{val})_t\}$         $\triangleright$ according to Eqn. (3)

        Clip $\widetilde{p}_{r,t}^{z=y}$ to be within $(\frac{1}{Z}, 1]$

        Update $\epsilon_{r,t} \leftarrow (\widetilde{p}_{r,t}^{z=y} - 1) \cdot \frac{Z}{1-Z}$         $\triangleright$ according to Eqn. (4)

        Update $\widetilde{p}_{r,t} \leftarrow p_r(1 - \epsilon_{r,t}) + \frac{\epsilon_{r,t}}{Z}$         $\triangleright$ according to Eqn. (1)

    **end for**

**end for**

---

$$\widetilde{p}_r^{z=y} = 1 - \epsilon_r + \frac{\epsilon_r}{Z}. \tag{2}$$

Since well-calibrated uncertainty estimates should be aligned with the empirical accuracy, we use the calibration performance in the *validation set* to help update $\widetilde{p}_r^{z=y}$ for the *training data*. Specifically, we first rank the adversarial robustness of the validation data and split the validation set into $R$ equally-sized subsets. Then, we use the difference between confidence and accuracy in the $r$-th adversarial robustness validation subset $\mathrm{conf}(S_r^{val}) - \mathrm{acc}(S_r^{val})$ to update the soft label for the correct class of training data in the $r$-th adversarial robustness training subset $S_r^{train}$,

$$\widetilde{p}_{r,t+1}^{z=y} = \widetilde{p}_{r,t}^{z=y} - \alpha \cdot \{\mathrm{conf}(S_r^{val})_t - \mathrm{acc}(S_r^{val})_t\} \tag{3}$$

where $\widetilde{p}_{r,t}^{z=y}$ is the soft label of the correct class in the $r$-th adversarial robustness training subset at time step $t$. The accuracy and the confidence of $S_r^{val}$ are defined as $\mathrm{acc}(S_r^{val}) = \frac{1}{|S_r^{val}|}\sum_{i \in S_r^{val}} \mathbf{1}(\hat{y}_i = y_i)$ and $\mathrm{conf}(S_r^{val}) = \frac{1}{|S_r^{val}|}\sum_{i \in S_r^{val}} \hat{p}_i^{z=\hat{y}_i}$, where $\hat{y}$ and $y$ is the predicted class and the true class respectively, $\hat{p}^{z=\hat{y}}$ denotes the the predicted probability of the predicted class. The hyperparameter $\alpha > 0$ plays a role as a learning rate to update the soft label $\widetilde{p}_{r,t}^{z=y}$ based on the difference between the predicted confidence and accuracy in the valiadation set. Intuitively, if we assign a large $\widetilde{p}_r^{z=y}$ to training data, then the network tends to make a high confidence prediction and vice versa. Therefore, if the confidence is greater than the accuracy ($\mathrm{conf}(S_r^{val}) > \mathrm{acc}(S_r^{val})$)) in the validation set, we should reduce $\widetilde{p}_r^{z=y}$ to teach the network to be less confident. Otherwise, we should increase $\widetilde{p}_r^{z=y}$. In addition, we also need to constrain $\widetilde{p}_r^{z=y}$ to be within $(\frac{1}{Z}, 1]$ after each update as it stands for the true probability of the correct class, where $Z$ is the number of classes in the dataset.

For a given $\widetilde{p}_r^{z=y}$, we can easily obtain $\epsilon_r$ by reversing Eqn. (2):

$$\epsilon_r = (\widetilde{p}_r^{z=y} - 1) \cdot \frac{Z}{1-Z}, \tag{4}$$

and the soft labels for all the classes $\widetilde{p}_r$ can be computed according to Eqn. (1). We update the soft labels after each training epoch in our experiments.

Note that this adaptive learning mechanism can be easily applied to standard label smoothing without adversarial robustness slicing ($R = 1$). In this case, we can replace sweeping the hyperparameter $\epsilon$ with this adaptive learning method, named as "**Ada**ptive **L**abel **S**moothing" (**AdaLS**). Our proposed `AdaLS` and `AR-AdaLS` does not increase the inference time: we test `AdaLS` and `AR-AdaLS` exactly the same as a vanilla model.

## 5 EXPERIMENTS

We now test our methods on both clean and shifted data for CIFAR-10, CIFAR-100 and ImageNet for calibration. The shifted dataset (Hendrycks & Dietterich, 2019) consists of different types (19 types for CIFAR-10, 17 types for CIFAR-100 and 15 types for ImageNet) of corruptions, e.g., noise, blur, weather and digital categories that are frequently encountered in natural images. Each type of corruption has five levels of shift intensity, with higher levels having more corruption. For each shift intensity, we report the results with a box plot summarizing the 25th, 50th, 75th quartiles across the

Table 2: Ablation study of `AR-AdaLS` on CIFAR-100 and CIFAR100-C (corrupted). We report both accuracy and expected calibration error, denoted by **Acc** and **ECE** for the clean test set, and **cAcc** and **cECE** for CIFAR100-C. Arrow indicates the better direction; best calibration of single model is **bolded**.

| Method | Vanilla | Label Smoothing | Temperature Scaling | AR-AdaLS (pre-compute) | AR-AdaLS (on-the-fly) | Ensemble of Vanilla |
|---|---|---|---|---|---|---|
| **Acc/cAcc** ($\uparrow$) | 79.2/52.0 | 78.9/51.7 | 79.2/52.0 | 79.3/52.2 | 79.2/52.1 | 81.8/55.1 |
| **ECE/cECE** ($\downarrow$) | 6.1/18.2 | 2.8/16.3 | 4.3/14.0 | 2.6/14.2 | **2.3/13.2** | 2.2/10.5 |

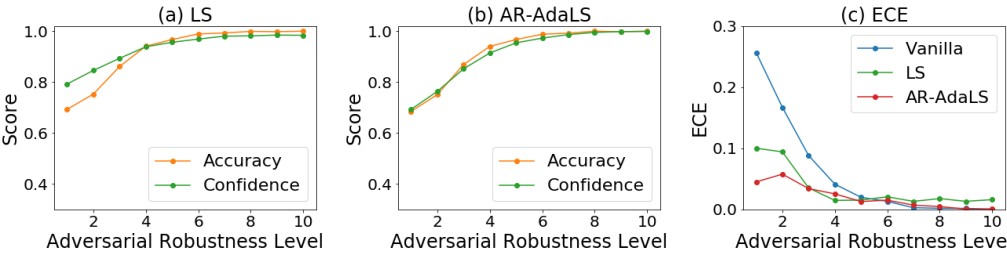

Figure 2: Comparison between LS and our `AR-AdaLS` on the clean test set of CIFAR-10. **(a)** and **(b):** Accuracy and confidence score of the predicted class in each adversarial robustness subset. **(c):** ECE of Vanilla, LS and `AR-AdaLS`.

types of shift. We first test single-model approaches and then explore how our methods compose with deep ensembles (Lakshminarayanan et al., 2017).

## 5.1 BASELINES

We compare our proposed **AR-AdaLS** with the following methods: Vanilla model that is trained with one-hot labels, Temperature Scaling (Guo et al., 2017), label smoothing (**LS**) (Szegedy et al., 2016) that softs labels by sweeping the hyperparameter $\epsilon$ which controls the smoothing degree in a range to find the best hyperparameter $\epsilon$, Adaptive Label Smoothing (**AdaLS**): we use our proposed adaptive learning mechanism introduced in Section 4.1 to automatically learn the hyperparameter $\epsilon$ rather than sweeping to find the best $\epsilon$. In addition, we also report the results of **Ensemble of Vanilla** (Lakshminarayanan et al., 2017) with $M = 5$ vanilla models independently trained with random initialization. We will further discuss that our method is complementary to deep ensembles. All the methods are trained with the same network architectures and training hyperparameters: e.g., learning rate, batch size, number of training epochs, for fair comparison. Please refer to Appendix A for all the training details and hyperparameters.

## 5.2 IMPROVEMENTS OVER SINGLE MODEL

**Ablation Study**   There are two options to compute adversarial robustness. One is "on-the-fly": to keep creating adversarial attacks during training, which provides precise adversarial robustness ranking but at the cost of great computing time. The other is to "pre-compute" adversarial robustness by attacking a vanilla model. This is more efficient but at the sacrifice of the precision of adversarial robustness ranking. We perform experiments on CIFAR-100 as an example to compare the performance of `AR-AdaLS` based on the adversarial robustness that is "pre-computed" or "on-the-fly".

As shown in Table 2, generating adversarial robustness "on-the-fly" helps improve the calibration performance further for `AR-AdaLS` on both clean and shifted datasets compared to pre-computing adversarial robustness. When comparing `AR-AdaLS` with other single-model based methods, we can see that all models have similar accuracy on clean and shifted datasets. Both versions of `AR-AdaLS` significantly improve the calibration performance over label smoothing, especially under distributional shifts. Further, we observe that `AR-AdaLS` with pre-computed adversarial robustness has similar performance as Temperature Scaling but `AR-AdaLS` based on more precise adversarial robustness ("on-the-fly") is significantly better[2]. In the following sections, all results related to "AR-AdaLS" are based on pre-computed adversarial robustness for efficiency. This is because our main target

---

[2]Similarly, we find that pre-computed `AR-AdaLS` improves on LS and performs nearly as well as TS on CIFAR-10 and ImageNet. We also see similar improvements from on-the-fly `AR-AdaLS` on CIFAR-10, but we did not run on-the-fly `AR-AdaLS` for ImageNet due to the computational intensity.

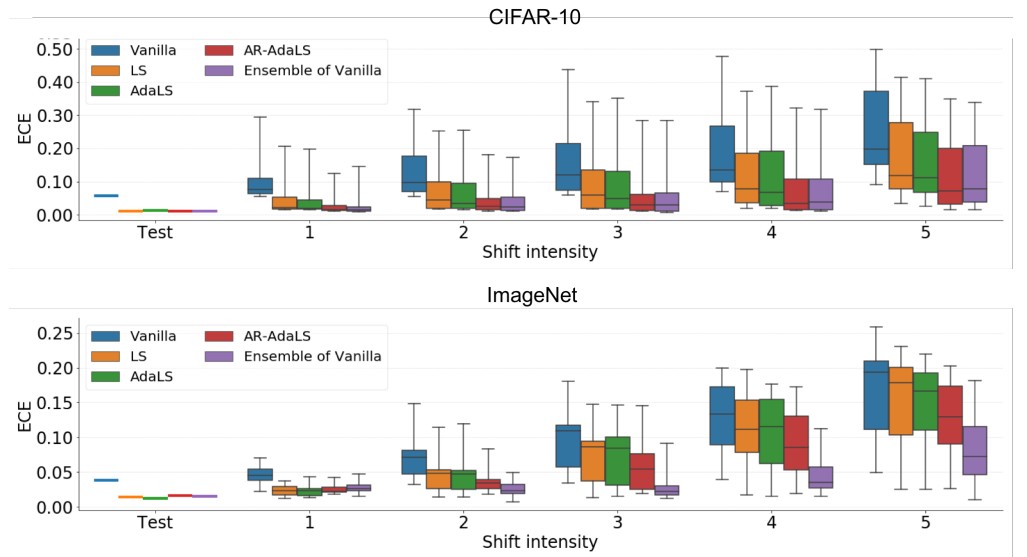

Figure 3: ECE on clean test and shifted data on CIFAR-10 and ImageNet. For each shift intensity, we show report a box plot summarizing the 25th, 50th, 75th quartiles across 19 shift types on CIFAR10-C and 15 shift types on ImageNet-C. The error bars indicate the $min$ and $max$ value across different shift types. Similar figures for Accuracy are shown in Figure 8 in Appendix and we observe that all the single-model based methods have comparable accuracy while ensembles achieve higher accuracy.

is to show that the idea of differentiating the training data based on their adversarial robustness is promising to improve model's calibration rather than pushing the results to the best.

**Visualization of improvement over label smoothing**   Since our proposed AR-AdaLS is built upon label smoothing (LS), in Figure 2 we visualize the effect of label smoothing (LS) and our AR-AdaLS. Comparing Figure 2 (a) and (b). AR-AdaLS is better at calibrating the data than label smoothing, especially on the unrobust examples (lower adversarial robustness level). Further, we show plots of ECE in Figure 2 (c). Both label smoothing and AR-AdaLS improve model's calibration over vanilla model and AR-AdaLS has the best performance among three methods. This suggests that AR-AdaLS is better at improving calibration in unrobust regions, not just on average.

**Generalization over shifted dataset**   Figure 3 and Table 4 summarizes the ECE for CIFAR-10 and ImageNet for both clean and shifted data with different levels of corruptions (Hendrycks & Dietterich, 2019). On the clean test set, all non-vanilla methods achieve comparable low values of ECE. When the intensity of shift increases, AR-AdaLS significantly outperforms other single-model based methods with the lowest ECE. Contrasting with LS and AdaLS, we see AR-AdaLS benefits greatly from the adversarial robustness slicing. As a result, our model learns to give smaller soft labels of the correct class to those adversarially unrobust training data, which can also be considered as outliers of the underlying data distribution (Carlini et al., 2019). Therefore, when tested on the shifted data that deep networks have been shown to produce pathologically over-confident predictions (Hendrycks & Dietterich, 2019), our model correctly learns to make a relatively lower confidence prediction, resulting in a better calibration performance. When we compare to an ensemble of *five* Vanilla models, we can see that AR-AdaLS achieves comparable calibration performance on CIFAR-10 and the ensemble is better under highly shifted data on ImageNet.

**Sensitivity analysis**   To analyze the effect of the number of adversarial robustness subset $R$ in AR-AdaLS, we plot the calibration error of AR-AdaLS with a varying $R$ on the clean CIFAR-10 and corrupted CIFAR-10-C in Figure 4. We can see that there is a significant drop in calibration error (ECE) when we increase the number of adversarial robustness subset $R$ from 1, where $R = 1$ denotes AdaLS. Further, the calibration error is robust when $R$ is chosen within the range [4, 16].

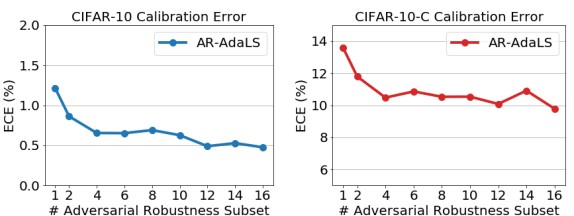

Figure 4: ECE on CIFAR-10 and CIFAR-10-C of AR-AdaLS with varying number of adversarial robustness subset $R$.

Figure 5: Histogram of predictive entropy on out-of-distribution data. Each model is trained on CIFAR-10 and tested on CIFAR-100.

Table 3: Mean of variance ($\times 10^{-2}$) across 19 types of shift for CIFAR-10-C and 15 types of shift for ImageNet-C. The best model is shown in **bold**.

| Dataset | CIFAR-10-C | | | | | ImageNet-C | | | | |
|---|---|---|---|---|---|---|---|---|---|---|
| Shift Intensity | 1 | 2 | 3 | 4 | 5 | 1 | 2 | 3 | 4 | 5 |
| Vanilla | 7.85 | 9.69 | 11.2 | 13.1 | 16.0 | 5.28 | 6.39 | 7.37 | 8.23 | 8.29 |
| LS | 5.54 | 6.95 | 8.11 | 9.65 | 11.8 | 4.86 | 5.84 | 6.78 | 7.55 | 7.41 |
| AdaLS | 5.47 | 6.87 | 7.95 | 9.44 | 11.5 | 4.79 | 5.77 | 6.66 | 7.51 | 7.56 |
| AR-AdaLS | **4.21** | **5.06** | **5.73** | **6.66** | **8.24** | **4.53** | **5.49** | **6.12** | **6.76** | **6.66** |

**Improvements on Out-of-Distribution Data**   We further study the performance of `AR-AdaLS` when predicting on out-of-distribution (OOD) data. In Figure 5, we compare the performance of Vanilla, Label Smoothing and `AR-AdaLS` by plotting the histogram of the entropy on the OOD data (higher entropy on OOD is better). Each model is trained on CIFAR-10 dataset and then tested on CIFAR-100 dataset. We can clearly see that `AR-AdaLS` significantly reduces the number of low-entropy prediction on OOD data, which demonstrates the effectiveness of `AR-AdaLS` even on fully out-of-distribution data.

**Improvements over stability**   Since we observe in Figure 1 that the most adversarially unrobust data are also very unstable, we test `AR-AdaLS` to see if it can help improve model stability, which is of great value in practice where high variance of a model is bad for churn and deploying ensembles is too costly (Milani Fard et al., 2016). Experiments show that `AR-AdaLS` can effectively reduce the variance of a model compared to a vanilla model and label smoothing on CIFAR-10 and ImageNet. An extensive set of results are shown in Table 3 and Figure 11 in the Appendix.

### 5.3   COMBINATION WITH DEEP ENSEMBLES

We now discuss if the two best models, deep ensembles and our `AR-AdaLS`, are complementary. To this end, we propose the following two ways to combine them: **Ensemble of `AR-AdaLS`:** As in Lakshminarayanan et al. (2017); Lee et al. (2015), we ensemble `AR-AdaLS` by training multiple independent `AR-AdaLS` models with random initialization, and average their predictions at inference.

**`AR-AdaLS` of Ensemble:** Instead of computing soft labels independently for each `AR-AdaLS`, we perform `AR-AdaLS` on the ensembled predictions, i.e., in Eqn (3) we compute confidence and accuracy based on the average of $M = 5$ model predictions. Each model is then supervised with the same soft labels. We will see this slight distinction in training is quite important.

**Discussion**   We present the results for CIFAR-10 and ImageNet in Figure 6 and Table 4. At a high level, we see that `AR-AdaLS` of Ensemble performs the best across both clean test data and all intensities of shifted data. Looking more closely, some trends emerge: all of the ensemble methods perform relatively well for highly shifted data (intensity 4–5), but Ensemble of `AR-AdaLS` performs much worse on less shifted and clean test data. Please refer to Figure 8 in Appendix for more extensive results on CIFAR-10 and Imagenet.

Digging deeper, we display the confidence of the predictd class and accuracy of each single model and the corresponding ensemble models on the clean test set of CIFAR-10 and ImageNet in Figure 10 in Appendix. We can clearly see that the ensemble models generally increase accuracy and decrease confidence compared to a single model, which results from the disagreement of the prediction of each single model in ensembles. Therefore, naive deep ensembles can improve calibration on highly shifted data where single-model is over-confident but can harm calibration if applied to a well-

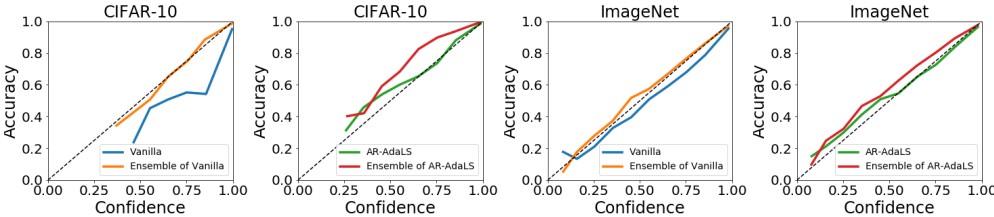

Figure 6: Comparison of ensembled models: ECE on both clean test data and shifted data on CIFAR-10. For each intensity of shift, we report a box plot summarizing the 25th, 50th, 75th quartiles across 19 types of shift in CIFAR10-C. The error bars indicate the min and max value across shift types.

Table 4: Mean of ECE ($\times 10^{-2}$) across 19 types of shift for CIFAR-10-C and 15 types of shift for ImageNet-C. The best single model and ensemble model are shown in **bold** respectively.

| Dataset | CIFAR-10-C | | | | | ImageNet-C | | | | |
|---|---|---|---|---|---|---|---|---|---|---|
| Shift Intensity | 1 | 2 | 3 | 4 | 5 | 1 | 2 | 3 | 4 | 5 |
| Vanilla | 9.59 | 12.9 | 15.6 | 19.5 | 25.7 | 4.67 | 6.94 | 9.60 | 13.2 | 16.6 |
| LS | 4.45 | 7.06 | 9.15 | 12.1 | 17.5 | 2.38 | 4.62 | 7.39 | 11.4 | 15.1 |
| AdaLS | 4.06 | 6.65 | 8.85 | 11.7 | 16.5 | **2.37** | 4.48 | 7.18 | 11.1 | 14.6 |
| AR-AdaLS | **2.57** | **4.09** | **5.64** | **7.85** | **11.9** | 2.44 | **3.58** | **5.80** | **9.33** | **12.9** |
| Ensemble of Vanilla | 2.47 | 4.13 | 5.73 | 8.03 | 12.1 | 2.77 | **2.51** | 2.87 | 4.64 | 8.26 |
| Ensemble of LS | 2.77 | 3.24 | 3.96 | 5.20 | 7.85 | 4.96 | 4.18 | 3.65 | 3.65 | 6.94 |
| Ensemble of AdaLS | 3.48 | 3.74 | 4.86 | 5.95 | 7.74 | 5.20 | 4.36 | 3.61 | 3.74 | 6.92 |
| Ensemble of AR-AdaLS | 4.23 | 4.52 | 5.23 | 5.68 | **7.68** | 5.76 | 5.23 | 4.51 | 3.69 | **6.33** |
| AR-AdaLS of Ensemble | **1.97** | **2.98** | **3.91** | **5.38** | 7.98 | 3.41 | 3.03 | **2.84** | **3.61** | 7.03 |

Figure 7: Reliability diagram of accuracy versus confidence of single model and ensemble model on the clean test of CIFAR-10 and ImageNet. The perfect calibrated model should be aligned with the diagonal dotted line (above is under-confident, below is over-confident).

calibrated single-model. This is made clearer in Figure 7: while deep ensembles make over-confident vanilla model well calibrate, it leads the well calibrated AR-AdaLS models to be under-confident (similar patterns observed on LS and AdaLS, shown in Figure 9 in Appendix). From this perspective, AR-AdaLS of Ensemble avoids this issue by adaptively adjusting smoothing to keep the ensemble well calibrated. Taken together we see that AR-AdaLS improves calibration for both single-models and ensembles.

## 6 CONCLUSION

In this paper, we have explored the correlations between adversarial robustness and calibration. We find across four datasets that adversarially unrobust (easily attacked) data are more likely to have poorly calibrated and unstable predictions. Based on this insight, we propose AR-AdaLS to adaptively smooth the labels of the training data based on their adversarial robustness. In our experiments we see that AR-AdaLS is more effective than previous label smoothing methods in improving calibration, particularly for shifted data, and can offer improvements on top of already strong ensembling methods. We believe this is an exciting new use for adversarial robustness as a means to more generally improve model trustworthiness, not just by limiting adversarial attacks but also improving uncertainty on unexpected data. We hope this spurs further work at the intersection of these areas of research.

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

## A  IMPLEMENTATION DETAILS

### A.1  SVHN

We use a simple network architecture with 7 convolutional layers and follow all the training details introduced in Qin et al. (2020a) for SVHN. This network architecture achieves the state-of-the-art accuracy on SVHN.

### A.2  CIFAR-10

All the experimental results on CIFAR-10 were obtained with a ResNet-29 v2 (He et al., 2016b) with a batch size of 256. The network is trained with Adam optimizer (Kingma et al., 2015) for 200 epochs. The initial learning rate is $10^{-3}$ and decayed down to $10^{-4}$ after 80 epochs, $10^{-5}$ after 120 epochs, $10^{-6}$ after 160 epochs and $0.5 \times 10^{-6}$ after 180 epochs. We adapted the following data augmentation and training script at https://keras.io/examples/cifar10_resnet/. The training mechanism is the same for all the methods that we compare in the main paper. We randomly split the training dataset into training data of 45000 images and 5000 images as the validation set. The test set has 10000 images.

For label smoothing (LS), we sweep the hyperparameter $\epsilon$ within the range [0, 0.1] with a step size 0.01 and find that the network has the best calibration performance on the validation set when $\epsilon = 0.02$.

For Adaptive Label Smoothing (AdaLS), there is a hyperparameter $\alpha$ which plays a role as learning rate in the adaptive learning mechanism. We choose hyperparameter $\alpha$ based on the calibration performance on the validation set. Specifically, we run experiments with $\alpha \in \{0.005, 0.01, 0.05, 0.1\}$ and find that $\alpha = 0.05$ achieve the best calibration performance.

Similarly, for Adversarial Robustness based Adaptive Label Smoothing (AR-AdaLS), we choose the hyperparameter $\alpha$ from the set $\{0.005, 0.01, 0.05\}$ and empirically set $\alpha = 0.005$ which has the best calibration performance on the validation set. We use the same hyperparameter $\alpha = 0.005$ without further tuning for AR-AdaLS of Ensemble.

All the results of ensemble models are obtained via training 5 independent models with random initializations.

### A.3  CIFAR-100

We train a Wide ResNet-28-10 v2 (Zagoruyko & Komodakis, 2016) to obtain the state-of-the-art accuracy for CIFAR-100. We adapt the same training details and data augmentation at https://github.com/google/edward2/blob/master/baselines/cifar/deterministic.py.

For label smoothing, we e sweep the hyperparameter $\epsilon$ within the range [0, 0.1] with a step size 0.01 and find that the network has the best calibration performance on the validation set when $\epsilon = 0.07$.

The hyperparameter $\alpha$ is set to be 0.005 in `AR-AdaLS`. For `AR-AdaLS` that generated with on-the-fly adversarial examples, we recompute the adversarial robustness for training and validation sets after 65, 130 epochs.

### A.4 IMAGENET

All the experiments on ImageNet were obtained via training a ResNet-101 v1 (He et al., 2016a) following the training script at `https://github.com/google/edward2/blob/master/baselines/imagenet/deterministic.py`. The network is trained with a batch size of 128 for each TPU core with SGD optimizer for 90 epochs. The input image is normalized (divided by 255) to be within [0,1]. We randomly divide 50000 validation images into validation set with 25000 images and test set with 25000 images. Note that the same dataset and training mechanisms are used for all the methods that we compare in the main paper.

For Label Smoothing (LS), we sweep the hyperparameter $\epsilon$ within the range [0, 0.1] with a step size 0.01 and find that the best calibration performance on the validation set is achieved by setting $\epsilon = 0.02$.

For Adaptive Label Smoothing (AdaLS), we sweep the hyperparameter $\alpha$ in the set $\{0.005, 0.01, 0.03, 0.05, 0.1\}$ and set it to be $\alpha = 0.03$ for the best calibration performance on the validation set.

We empirically set $\alpha = 0.001$ for `AR-AdaLS` in the first 60 epochs of the training and then increase it to 0.05 for the next 30 epochs. The same hyperparameter $\alpha$ is used for `AR-AdaLS` of Ensemble without further tuning.

All the ensemble models are a combination of 5 independent models with random initializations.

### A.5 CW ATTACKS

To compute the adversarial robustness, we construct $\ell_2$ based CW attacks (Carlini & Wagner, 2017b) following the code at `https://github.com/tensorflow/cleverhans/blob/master/cleverhans/attacks/carlini_wagner_l2.py`. Specifically, we set the binary search steps to be 3, max iterations to be 500 and learning rate to be 0.005. The generated untargeted CW attacks can achieve 100% success rate for all the datasets that we consider: SVHN, CIFAR-10, CIFAR-100 and ImageNet. We set the number of adversarial robustness training subset and validation subset to be $R = 10$ respectively.

## B CALIBRATION PERFORMANCE: ADDITIONAL RESULTS

In Figure 8, we show ECE and accuracy of all the single-models and their corresponding ensembles on the clean test and shifted CIFAR-10 and ImageNet. The high-level conclusion is: 1) all the ensemble models that we compare have similar accuracies, which are higher than single-models. 2) all the ensemble methods perform relatively well for highly shifted data but ensembles of well calibrated models (Ensemble of LS, Ensemble of AdaLS, Ensemble of `AR-AdaLS`) perform much worse on less shifted and clean test data. 3) `AR-AdaLS` of Ensemble successfully learns to adaptively adjust its smoothing level to keep the ensemble well calibrated on both clean and shifted dataset.

Figure 9 is to validate the observation that naive ensembling of a single model can improve calibration on highly shifted data where single-model is over-confident but can harm calibration on clean test data where single model is well calibrated. As shown in Figure 9, we can see that if the single-model is over-confident (Vanilla), Ensemble of Vanilla greatly improves the model's calibration. However, when the single model is well-calibrated (LS, AdaLS, `AR-AdaLS`), naive ensembling of these models leads to an under-confident model with a worse calibration performance. In contrast, `AR-AdaLS` of Ensemble successfully avoids this issue by adaptively smoothing the training labels according to adversarial robustness of training data.

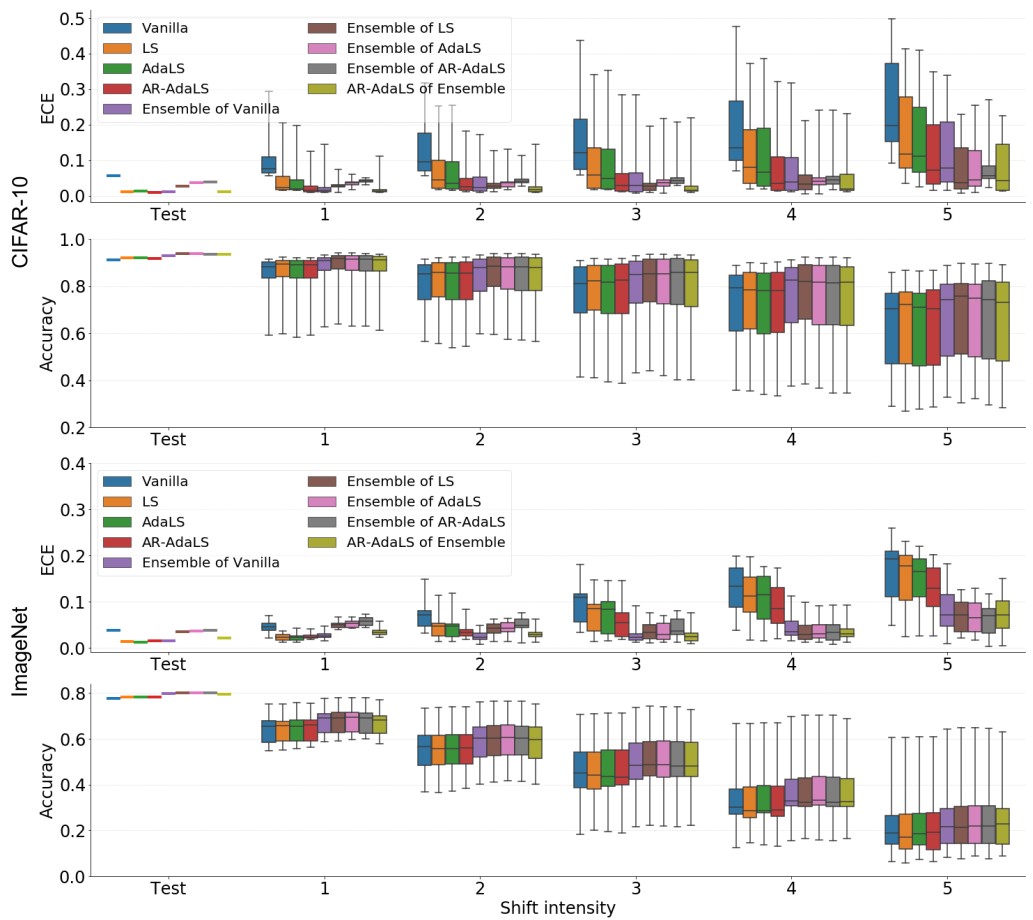

Figure 8: Comparison of ensembled models: ECE and Accuracy on both clean test data and shifted data on CIFAR-10 and ImageNet. For each intensity of shift, we show the results with a box plot summarizing the 25th, 50th, 75th quartiles across 19 types of shift on CIFAR-10-C and 15 types of shift on ImageNet-C. The error bars indicate the $min$ and $max$ value across different shift types.

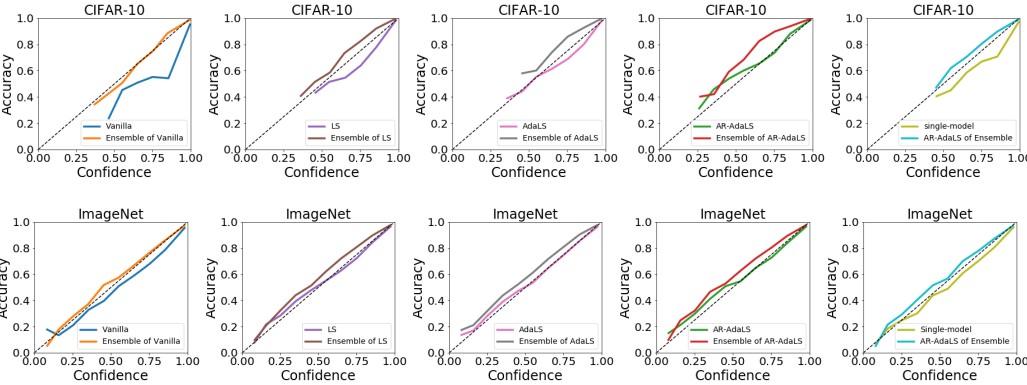

Figure 9: Reliability diagram of accuracy versus confidence of single model and ensemble model on the clean test of CIFAR-10 and ImageNet. The perfect calibrated model should be aligned with the diagonal dotted line (above is under-confident, below is over-confident).

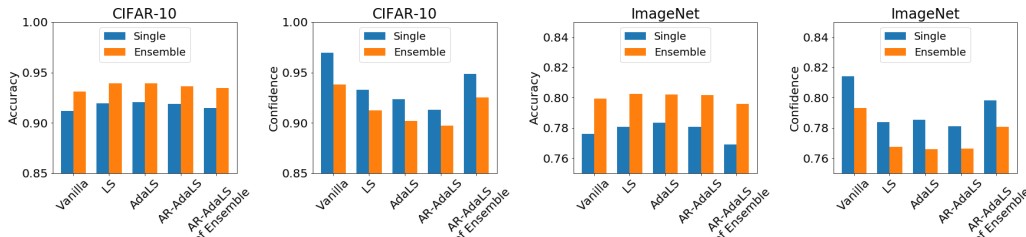

Figure 10: Comparing accuracy and confidence of the predicted class between single model and the corresponding ensemble model for each method.

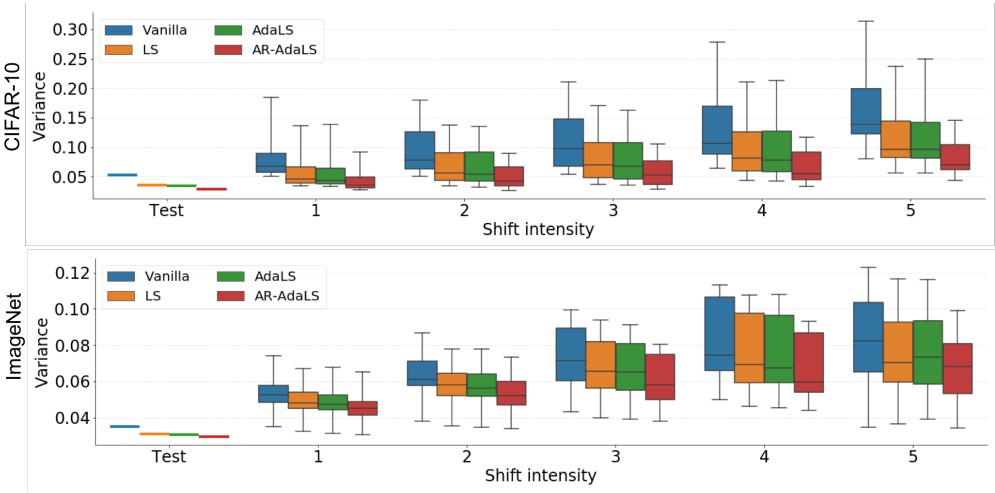

Figure 11: Variance on clean test and shifted data on CIFAR-10 and ImageNet. For each shift intensity, we show the results with a box plot summarizing the 25th, 50th, 75th quartiles across 19 shift types on CIFAR10-C and 15 shift types on ImageNet-C. The error bars indicate the $min$ and $max$ value across different shift types.

