# OpenReview forum: "Improving Calibration through the Relationship with Adversarial Robustness"
_ICLR.cc/2021/Conference — Reject_

### Official Review · AnonReviewer2 · 2020-10-22
**a useful connection and compelling results but not fully explored**

**Rating:** 7
**Confidence:** 3

**Review:**

Summary: This paper proposes a new method (AR-AdaLS) for label smoothing to improve deep network calibration. In particular, the authors draw a connection between lack of calibration (overconfidence) and examples which are prone to adversarial attacks. They show that by generating smoothed targets based on the adversarial robustness of an example, they can further improve model calibration beyond traditional label smoothing.

Pros: The paper demonstrates a clear connection between per-example calibration error (in general, overconfidence) and a lack of adversarial robustness.

Method and results are clear and seem to be well situated in the literature.

They demonstrate the outperformance of their method relative to Label Smoothing in calibrating across several datasets, model architectures, and domain shifts.

Cons: I'm not an expert in this field, but the novelty of the adversarial robustness - overconfidence connection is not entirely clear to me. Muller et al. describe the behavior of label smoothing as necessarily reducing the difference between logits of two classes and thereby the confidence of the prediction, particularly for outliers. This would seem to be the same effect you describe, but perhaps less explicitly.

The methods used as baselines, label smoothing and temperature scaling, are attractive for their simplicity and ease of training. While you note that your model has the same latency at inference, it seems important to know how much extra computation is required in training.

The improvement of AR-AdaLS relative to vanilla ensemble (and ensemble vanilla relative to AR-AdaLS of ensemble) is meaningfully different for ImageNet. You mention this may be due to larger dataset/less overfitting but it would be useful to better understand this - how do other forms of regularization relate to AR-AdaLS?

I find the discussion of Figure 5 somewhat unsatisfying - it's not entirely clear to me why averaging the predictions of 5 well-calibrated models results in a miscalibrated model

What is sensitivity to hyperparameter R?

Minor:

Complementary, not complimentary

In related work: multiclasss

---

> ### Author Response · Authors · 2020-11-24
> **To Reviewer2**
>
> Thanks very much for your support of our work and we would love to address your concerns as follows:
> 1. Q: Confusion of adversarial robustness-overconfidence connection.
> A: First, our AR-AdaLS is built on top of label smoothing. Therefore, the benefit of label smoothing introduced in Muller et al, that is to reduce the difference between logits of two classes, also holds true on AR-AdaLS. Second, the main difference between AR-AdaLS and label smoothing is that AR-AdaLS smooths the labels of the training data to **different** extents based on their adversarial robustness while standard label smoothing just smooths all the training labels to the **same** extent. Assigning different labels to different training data is highly motivated by the nice correlation we revealed between adversarial robustness and over-confident predictions. Specifically, we find that for those most unrobust data, their predictions are more likely to be overconfident; for those robust data, their predictions are more likely to be well-calibrated. Therefore, we need to assign a smaller confidence score of the labels to those unrobust training data.
>
> 2. Q: Extra computation:
> A: As you point out, AR-AdaLS shares the same latency as the vanilla model or label smoothing at inference, which we believe is more important while deploying a model in practice. Second, the extra computation mainly comes from computing the adversarial robustness of each training and validation data, which involves constructing CW attacks. Generating CW attacks once for both training data and validation data takes around an extra half of the training time. We believe it is an interesting further research question if this adversarial robustness of examples can be approximated more efficiently via other adversarial attacks.
>
> 3. Q: Discussion of Figure 5. A: Sorry for the confusion of Figure 5 (now Figure 7 in the new version) and we add more explanations in the main text. First, we analyze the effect of an ensemble model, which is to improve the accuracy and reduce the confidence resulting from the disagreement of the prediction of each single model in ensembles. This is validated by Figure 10 in Appendix, where we compare the accuracy and confidence of the predicted class between each single and the corresponding ensemble model for 5 different methods.  Therefore, the overall effect of deep ensembles is to drive a model to be less confident, demonstrated in Figure 7 that the lines of ensemble models move to the upper left. Based on this, when ensembles are applied to a well-calibrated method, the resultant model becomes under-confident and has a worse calibration performance. Further, that ensemble can hurt the calibration performance of data augmentation, e.g, mixup, is also observed in [1] and [2].
>
> 4. Q: Sensitivity analysis of hyperparameter R. A: We perform a sensitivity analysis of hyperparameter R and found that the performance is robust when R is chosen between 4 to 16. We add a detailed plot of ECE on CIFAR-10 and CIFAR-10-C of AR-AdaLS
> with varying number of adversarial robustness subset R in Figure 4. In addition, we also include a discussion about sensitivity analysis on the bottom of Page 8 in the main text.
>
> [1] Wen et al, 2020. Combining Ensembles and Data Augmentation can Harm your Calibration
>
> [2] Rahaman, R. and Thiery, A.H., 2020. Uncertainty Quantification and Deep Ensembles.

---

### Official Review · AnonReviewer4 · 2020-10-26
**Good motivation, some parts remain unclear**

**Rating:** 5
**Confidence:** 3

**Review:**

### Main contribution
1. The author find that unrobust data may cause worse calibration.
2. This work propose to encourage models to be adversarial robust to help calibration with a modified version of label smoothing: AR-AdaLS

### Clarity
1. The introduction of the backgrounds for uncertainty estimates is not very clear
2. The proposal of AR-AdaLS is clear and reasonable
3. The argument of "that unrobust data may cause worse calibration" is not fully demonstrated in Sec 3.

### Questions
1. In adversarial learning area, people usually refer "robustness" for models rather than data. We can say a model is robust or not. But it is a little bit confusing when you apply the term on data. Can the author provide references of other canonical works that also use this phrase?
2. In Sec 3, the author stated they consider adversarial perturbation δ as a measure of robustness. Then why bother to divide the data into ten sets based on δ in your most import illustrations, which is Fig 1. What will happen if you directly present the relationship between δ and score? Does the 10th set represents the most robust data or does the 1st set do?
3. What conclusion can we draw from the first row of Fig 1? How does it relate to the argument that "the model is sensitive to small perturbations are more likely to have poorly calibrated predictions"?
4. I am familiar with the area of adversarial robustness and out-of-distribution detection, but not very confident about uncertainty estimates. I also did not find any specific introduction or helpful information in Sec 2. It seems to me that this area is not very popular and you can only compare your method with a few of other works, most of which are not even proposed for calibration in the first place (Label Smoothing/Mixup/Temperature Scaling). So can the author illustrate how the area of CALIBRATION differs from OOD detection? What is the most important evaluation metric for this problem? And maybe tell me which one is the most canonical work in this area so that I can refer to it as a comparison.

I will raise my score if the author can fully address my questions.

---

> ### Author Response · Authors · 2020-11-24
> **To Reviewer4**
>
> Thanks very much for your reviews and we answer your questions as follows:
>
> 1.Q: Definition of adversarial robustness.
> A: We agree that calling models robust is more common, but measuring the robustness of examples has been discussed in a few recent works. As we mentioned in related work “Recently, Carlini et al. (2019); Stock & Cisse´ (2018) define adversarial robustness as the minimum distance in the input domain required to change the model’s output prediction.”  In addition, we also emphasize it in the definition of adversarial robustness in Section 3. Please refer to [1] & [2], which inspired us to use “adversarial robustness” to represent the robustness of data.
>
> [1] Nicholas Carlini et al. Distribution density, tails, and outliers in machine learning: Metrics and applications. ArXiv 2019.
>
> [2] Pierre Stock & Moustapha Cisse. Convnets and imagenet beyond accuracy: Understanding mistakes and uncovering biases.
>  ECCV 2018.
>
> 2.Q: Why do we need to divide the data into subsets?
> A: Accuracy and ECE are both most meaningful to compute over sets of data "for statistical significance". For example, we can't use the accuracy of one data point since it'll be either 0 or 1, and it doesn't make sense to smooth the confidence to 0 if a data point is mis-classified. Therefore, we cannot present the relationship between $\delta$ for each data point and its corresponding calibration score (ECE) (which takes accuracy as an input and thus cannot be computed for each data point).
> The 10th set represents the most robust data as mentioned in the title of Figure 1.
>
> 3.Q: Conclusion from the top row of Figure 1?
> A:  First, we want to point out that a model’s calibration measures the alignment between a model’s accuracy and confidence. A well-calibrated model means that the predicted confidence is aligned with the predicted accuracy. For example, if a model’s accuracy is around 60%, then if the model’s predicted confidence is 0.6, that means the model is well-calibrated; if the model’s predicted confidence is greater than 0.6, that means the model is over-confident; if the model’s predicted confidence is smaller than 0.6, that means the model is under-confident. As shown in the top row of Figure 1, we plot both the accuracy and confidence for each adversarial robustness subset. We can see that the gap between accuracy and confidence is not always the same for all the adversarial robustness subsets. For the test data with small adversarial robustness (with small adversarial robustness level), the confidence is significantly higher than the accuracy. However, for the robust test data (with high adversarial robustness level), confidence is well aligned with the accuracy. Therefore, we draw a conclusion at the end of the second paragraph in Section 3.1 Correlations, “This indicates that although vanilla classification models achieve the state-of-the-art accuracy, they tend to give over-confident predictions, especially for those unrobust data.”
>
> 4.Q: Difference between Calibration and OOD detection.
> A: As we mentioned above, a model’s calibration performance measures the alignment between a model’s accuracy and confidence. The predicted confidence of a well-calibrated model can tell us how much we should trust the model’s prediction. Calibration can be tested both on clean test data or on the shifted test data (e.g., we test the model’s calibration performance on the corrupted datasets, CIFAR10-C, CIFAR100-C and ImageNet-C. The corrupted datasets are constructed by applying different types of corruptions, e.g., noise, blur, weather and digital categories, to the clean test data). Instead, OOD detection is mainly focusing on detecting out-of-distribution data. A common way to test the performance of OOD detection is to train a model on one dataset, e.g., CIFAR-10,  and then test it on another dataset including different classes in the training dataset, e.g., CIFAR-100.
>
> In addition, we include OOD experiments in Figure 5 to show the effectiveness of AR-AdaLS even on fully OOD data by significantly reducing the number of low-entropy prediction on OOD data (see Section "Improvements on Out-of-Distribution Data" on page 8).
>
> The most important evaluation metric for calibration is ECE (expected calibration error), which is used in our work. We would like to refer to the following two works as “canonical” in calibration similar to our problem setting. The first one is “On calibration of modern neural networks. ICML 2017”, and another one is “Can you trust your model’s uncertainty? evaluating predictive uncertainty under dataset shift. NeurIPS, 2019.” The second paper is a benchmark paper which evaluates many calibration methods under data shift and concludes that deep ensembles have the best calibration performance. That is why we compare our method with ensembles and combine AR-AdaLS with ensembles to further improve the calibration performance.

---

### Official Review · AnonReviewer1 · 2020-10-28
**Missing baselines, incomplete experiments, only small and unclear benefits**

**Rating:** 2
**Confidence:** 5

**Review:**

The authors propose an extension to label smoothing, where the smoothing parameter is determined based on the miscalibration of the validation set. They then compare the performance of their method in terms of calibration under domain shift to a small set of baselines.

I have 3 main concerns regarding missing baselines, missing experiments and only marginal benefits over state-of-the-art.

My first main concern is that the authors limit their comparison to rather dated baselines (2018 and older), when there has been a lot of active research in this field in the past year. In particular, I am disappointed that while the authors cite recent work on MixUp showing its benefits for calibration (Thulasidasan et al., NeurIPS 2019) they do not include MixUp as baseline (where also inputs are smoothed rather than labels only). Other notable recent work that should be included as baseline is Verified Uncertainty Calibration, Kumar et al., NeurIPS 2019, which has been shown to be superior to Temperature Scaling.  In addition, a baseline which also exploits links between calibration and adversarial, Stutz et al., ICML 2020, should be included as prior work.
Another aspect of the presented work is the exploration of constructing an ensemble of neural nets trained with label smoothing. In light of this, it is crucial to also compare the presented approach to Mix-n-Match, Zhang et al., ICML 2020, who present work on ensemble methods for uncertainty calibration using label smoothing.

 My other main concern is a whole set of missing experiments investigating calibration in truly OOD scenarios. Snook et al, on whose work this paper heavily builds, point out that in addition to the domain drift scenarios explored by the authors, it is crucial to investigate performance in truly OOD scenarios, where the test data is drawn from a distribution far a way from the training distribution. Since the model is per definition not able to make a correct prediction, in this scenario entropy is used to quantify model the quality of the predictive uncertainty. The authors should add these experiments for all datasets.
In this context, it would also interesting to investigate the quality of predictive uncertainty of OOD detection methods. While such comparisons are not always meaningful, a model strongly related to label smoothing/MixUp is Hendrycks et al., ICLR 2019, where a GAN is trained to learn OOD samples, where in MixUp inputs are smoothed to generate OOD samples. A comparison to this approach would also be interesting.

Finally, even though important baselines as well as experiments for truly OOD scenarios are missing, benefits over a simple ensemble of vanillas remain unclear. For large-scale data (i.e. Imagenet), ensemble of Vanilla perform consistently better than the proposed method in terms of ECE under domain shift. For CIFAR-10, performance is comparable to deep ensembles and only for AR-AdaLS of Ensemble for CIFAR-10 there is a marginal improvement for a subset of perturbation strengths. I would have liked to see also evaluation of ECE under domain shift for CIFAR-100 and SVHN. Also missing is an evaluation of AR-AdaLS of Ensemble for Imagenet. Finally, results for non-image data, e.g. using a recurrent architecture (as in Snoek et al) are missing.

Other concerns and open questions include:
Although the authors explain the hyperparameters they use in the appendix, a detailed sensitivity analysis is missing to understand how sensitive the method is with respect to alpha and R.

The authors use „Ensemble of Vanilla“ as baseline - however, Snoek et al. have shown that it is actually deep ensembles that perform best; in addition to the ensemble effect they are also trained using adversarials. What is the performance of actual deep ensembles rather than ensemble of vanilla?

A conceptual concern is that in the proposed approach the validation set becomes part of the training set since it is used during training to update epsilon and thus is not anymore the independent set that may be necessary to tune other hyperparameters/for early stopping; I wonder whether this data leak may lead to problems related to overfitting?

Please report a proper scoring rule in addition ECE (e.g. Brier score).

A minor point ist that different colors are used for AR-AdaLS  and Ensemble of Vanilla in figure 3 and figure 4.

Unfortunately the authors do not provide any code, which make reproducibility difficult.



####post rebuttal####
The contribution of this paper is marginal only. The link between adversarial robustness and calibration has been explored previously: Snoek et al. NeurIPS 2019 have shown that adversarial training as part of deep ensembles leads to better calibration under domain shift. Unfortunately the authors do not compare their approach to these deep ensembles, but only an ensemble of differently initialised vanilla networks without adversarial training, which they call deep ensembles (section 5.1). Also in terms of label smoothing the contribution is marginal: in their rebuttal the authors show that MixUp training - a different implementation of label smoothing combined with input smoothing - has a better performance than their method (ECE of 1.8 MixUp vs 2.3 their method for CIFAR-100); they do show that further post-processing improves their method, but this is likely true for MixUp too (results not shown). For ImageNet results for MixUP are not shown, nor for calibration under domain shift where MixUp is likely to perform well too.
 Taken together, this suggests that the link between adversarial robustness and calibration is mainly a link between OOD samples and calibration: generating OOD samples with input smoothing in MixUp works very well compared to the proposed approach, as does adversarial training in deep ensembles (both of which was shown in prior work). In summary, the proposed approach lacks novelty and performs worse than baselines for complex datasets.

Lack of code during the reviewing phase means it is not possible to review reproducibility of results.

---

> ### Author Response · Authors · 2020-11-24
> **Response to Reviewer1**
>
> 1.Missing baselines: Thanks for your reviews but we would like to emphasize that the main contribution of our work is to reveal the correlation between two different research areas: adversarial robustness and calibration, and further we show we can use this correlation to improve calibration. Our main focus is not to develop a single method or model with state-of-the-art calibration but to demonstrate the value of new research work at the intersection of these areas and inspire others in this direction. Therefore, we believe the most important baseline method to understand the value of this insight is label smoothing, which we built our method on top of; we show a significant improvement of our method over standard label smoothing.
>
> Further, we also follow your suggestion to compare mixup and AR-AdaLS and find that AR-AdaLS is better than mixup on ImageNet and comparable to mixup on CIFAR datasets, as shown in Table 1. In addition, since our method focuses on changing the label space, we observe that our method is complementary to many other state-of-the-art methods, e.g, ensembles (shown in the paper) and the post-processing calibration methods, e.g., scaling-binning method proposed in [2], as shown in Table 1: AR-AdaLS + ScaleBin.
>
> Table 1: ECE score (%) on the clean CIFAR10 and CIFAR100 dataset with WideResNet 28-10.
>
> | Dataset  | AR-AdaLS| AR-AdaLS + ScaleBin[2] | Mixup [1] | Mix-n-Match [4] | CCAT [3]  |
> |---|:---:|:----:|:---:|:---:| :---:|
> |CIFAR10| 0.6 | 0.1| 0.8| 1.0| 2.4|
> |CIFAR100| 2.3 |1.5| 1.8|2.8| 4.2|
>
> The result of Mix-n-Match [4] and CCAT [3] are also reported in Table 1. We want to emphasize the difference between CCAT [3] and ours. CCAT [3] focuses on improving a model’s adversarial robustness but our work mainly focuses on improving a model’s calibration performance. Second, CCAT [3] builds their method on top of adversarial training while our method AR-AdaLS trains the model only with the clean training data rather than training on the adversarial examples. Therefore, CCAT [3] has a much lower clean accuracy compared to our method, e.g., CCAT: 94.6% vs. AR-AdaLS: 95.6% on CIFAR-10 and CCAT 75.3% vs. AR-AdaLS 79.2% on CIFAR-100.
>
> [1] Thulasidasan, S., et al, On mixup training: Improved calibration and predictive uncertainty for deep neural networks. NeurIPS  2019.
>
> [2] Kumar, A., et al, Verified uncertainty calibration. NeurIPS 2019.
>
> [3] Stutz, D., et al, Confidence-calibrated adversarial training: Generalizing to unseen attacks. ICML 2020.
>
> [4] Zhang, J., et al, Mix-n-Match: Ensemble and Compositional Methods for Uncertainty Calibration in Deep Learning. ICML 2020.
>
> 2.OOD experiments:
> First, improving calibration and understanding its relationship with adversarial robustness is important independent of OOD applications. Many existing work focusing on calibration, e.g,  Verified uncertainty calibration. NeurIPS 2019 and Mix-n-Match, ICML 2020, do not include OOD experiments.
>
> Second, we agree with the reviewer that our work is inspired by the work of (Snoek et al, Neurips 2019), but the major contribution and novelty of our work is to reveal the correlation between adversarial robustness and calibration, which is not discussed in the work of Snoek et al.
>
> Although OOD is not our focus, we still take the reviewe suggestions to includes experiments on OOD. Specifically, we plot the histogram of predictive entropy on out-of-distribution data of Vanilla, Label Smoothing and our AR-AdaLS. Each model is trained on CIFAR-10 and tested on CIFAR-100. Experimental results show that AR-AdaLS significantly reduces the number of low-entropy prediction on OOD data (see detailed discussion in Section "Improvements on Out-of-Distribution Data" on page 8).
>
> 3.ECE under domain shifts for CIFAR-100 has originally been included in Table 2.
>
> The results of ensembles on ImageNet are originally included in the appendix due to the space limitation and we move them into the main text in Table 4 as the final version allows one extra page. A full bar plot of ensembles results on ImageNet is shown in Figure 8 in Appendix.
>
> Unfortunately, there is no existing corrupted dataset for SVHN, but we believe the results on three datasets demonstrate the effectiveness of our algorithms.
>
> 4.Sensitivity analysis: We perform a sensitivity analysis of hyperparameter R and found that the performance is robust when R is chosen between 4 to 16. We add a detailed plot of ECE on CIFAR-10 and CIFAR-10-C of AR-AdaLS with varying number of adversarial robustness subset R in Figure 4. In addition, we also include a discussion about sensitivity analysis at the bottom of Page 8 in the main text.
>
> 5.“Ensembles of Vanilla” are equivalent to “deep ensembles” as illustrated in Section 5.1 Baselines.
>
> 6.Overfitting: First, we do not include any early stopping while training the model. Second, we did not observe any overfitting problem of our algorithm.
>
> 7.Availability of the code: We will open-source the code once the work is accepted.

---

### Official Review · AnonReviewer3 · 2020-10-30
**Improving Calibration with Adversarial Information**

**Rating:** 6
**Confidence:** 4

**Review:**


The author addressed my concerns. I’ll keep the score 6.

=======================

Summary:

The paper studied the relationship between adversarial robustness and calibration, then use the findings to improve label smoothing method. An adaptive label smoothing method (AR-AdaLS) is proposed to improve the calibration performance. Combining AR-AdaLS and deep ensemble can further improve the performance of deep ensemble method.

Strength:

1. The idea is novel and easy to understand. Experiments show the positive relationship between high adversarial robustness and better calibration. This work connects two field of studies.
2. AR-AdaLS improves the performance of label smoothing. AR-AdaLS can also be used to improve deep ensemble.
3. Improving calibration of deep neural networks is an important task.

Weakness:

1. Comparing the performances of LS and AR-AdaLS on CIFAR100 and CIFAR100-c, the improvements are actually not significant on CIFAR100, but AR-AdaLS is more computationally expensive. AR-AdaLS (on-the-fly) generates adversarial examples during training. This is very computationally expensive and will restrict the ability of the method to scale up to large dataset.
2. AR-AdaLS does not perform better than deep ensemble, which is the state-of-the-art (claimed by the author).
3. Three datasets are used in the paper but it seems that not all datasets are used to compare the performances of proposed model and other baselines.


Clarity and Correctness:

The paper is well written and easy to follow. The experiments look convincing. Boxplots comparison of ECE on CIFAR10 and ImageNet are provided.

Reproducibility:

Details of training and pseudocode is given but code is not available.

Questions:

The comparison of performances of ensemble models on CIFAR10 is provided but the comparisons on CIFAR100 and ImageNet are not. Why is that?

Conclusion:

The idea is novel and interesting but the empirical performance is not outstanding. Overall, I'm ok to accept the paper. I like the idea of using adversarial robustness to improve calibration. This paper reveals some interesting connections between the two things.

---

> ### Author Response · Authors · 2020-11-24
> **Response to Reviewer3**
>
> We really appreciate that you noticed that our main contribution is revealing an interesting connection between two research areas and use this correlation to further improve calibration. We answer your questions as follows:
>
> 1. Computational cost: We agree that generating adversarial examples on-the-fly would be more expensive, as we mention in the paper. But comparing standard label smoothing and AR-AdaLS (pre-computed), we have already seen the benefit of AR-AdaLS (pre-computed) over standard label smoothing, even on the large-scale dataset, e.g., Imagenet. In addition, we want to emphasize that AR-AdaLS does not have any latency in inference, which is more important when we deploy the model in practice.
>
> 2. “AR-AdaLS does not perform better than deep ensembles”: First, deep ensembles would be very expensive in inference, which is one of the major limitations of deep ensembles. Second, if there are computational resources for deep ensembles, we show that AR-AdaLS is complementary to deep ensembles and can further improve the calibration performance of deep ensembles.
>
> 3. Missing results on other datasets: We are sorry about this confusion but actually we included all the results on ImageNet in the appendix due to the space limitation, and we have moved the numerical result on ImageNet to the main text in Table 4 since one extra page is allowed. A full bar plot is shown in Figure 8 in appendix on CIFAR10 and ImageNet. Further, the result on CIFAR100 of ensemble is also included in Table 2.

---

### Author Response · Authors · 2020-11-24
**General Response to All Reviewers**

1. The key contribution of this work is that we reveal the correlation between adversarial robustness and calibration, drawing a link between two different research areas. We believe this can spur further work at the intersection of these areas of research. Based on this correlation, we want to show that the idea of differentiating the training data based on their adversarial robustness is promising to improve model’s calibration rather than pushing the results to be the best.

2. Due to the space limitation, we did not include all the experiments (e.g., ensemble results on ImageNet) in the main text but instead including them in the appendix. Since one extra page is allowed for the final version, we have moved these results into the main text in Table 4. A full bar plot of ensemble result on ImageNet is in Figure 8 in Appendix.

3. We will open-source all the code if the paper is accepted.

---

### Decision · Program_Chairs · 2021-01-07
**Final Decision**

**Decision:**

Reject

**Comment:**

This paper improves calibration of neural networks by investing its connection to adversarial robustness. Two reviewers suggested acceptance, and two did rejection. As the authors and some reviewers highlighted, AC also agreed that the correlation between adversarial robustness and calibration is interesting to explore. However, as R1 pointed out, AC also thinks that the experimental results are not strong enough to meet the high standard of ICLR, e.g., Mixup often outperforms the proposed method (without further post-processing) and the proposed method does not outperform the deep ensemble (although deep ensemble is expensive and both method can be combined). Due to this, AC doubts whether adversarial robustness is indeed the best way to improve calibration (it can be useful though). Hence, AC recommends rejection.